# Csf1 from marrow adipogenic precursors is required for osteoclast formation and hematopoiesis in bone

Leilei Zhong[1], Jiawei Lu[1], Jiankang Fang[1], Lutian Yao[1], Wei Yu[1,2], Tao Gui[1,3], Michael Duffy[1], Nicholas Holdreith[4,5], Catherine A Bautista[1], Xiaobin Huang[6], Shovik Bandyopadhyay[7,8], Kai Tan[5,9], Chider Chen[6], Yongwon Choi[10], Jean X Jiang[11], Shuying Yang[12], Wei Tong[4,5], Nathanial Dyment[1], Ling Qin[1]*

[1]Department of Orthopaedic Surgery, Perelman School of Medicine, University of Pennsylvania, Philadelphia, United States; [2]Department of Orthopaedics, Union Hospital, Tongji Medical College, Huazhong University of Science and Technology, Wuhan, China; [3]Department of Bone and Joint Surgery, Institute of Orthopedic Diseases, The First Affiliated Hospital, Jinan University, Guangzhou, China; [4]Division of Hematology, Children's Hospital of Philadelphia, Philadelphia, United States; [5]Department of Pediatrics, Perelman School of Medicine, University of Pennsylvania, Philadelphia, United States; [6]Department of Oral and Maxillofacial Surgery/Pharmacology, School of Dental Medicine, University of Pennsylvania, Philadelphia, United States; [7]Graduate Group in Cell and Molecular Biology, Perelman School of Medicine, University of Pennsylvania, Philadelphia, United States; [8]Medical Scientist Training Program, Perelman School of Medicine, University of Pennsylvania, Philadelphia, United States; [9]Center for Childhood Cancer Research, The Children's Hospital of Philadelphia, Philadelphia, United States; [10]Department of Pathology and Laboratory Medicine, Perelman School of Medicine, University of Pennsylvania, Philadelphia, United States; [11]Department of Biochemistry and Structural Biology, University of Texas Health Science Center at San Antonio, San Antonio, United States; [12]Department of Basic and Translational Sciences, School of Dental Medicine, University of Pennsylvania, Philadelphia, United States

*For correspondence: qinling@pennmedicine.upenn.edu

**Abstract** Colony-stimulating factor 1 (Csf1) is an essential growth factor for osteoclast progenitors and an important regulator for bone resorption. It remains elusive which mesenchymal cells synthesize Csf1 to stimulate osteoclastogenesis. We recently identified a novel mesenchymal cell population, marrow adipogenic lineage precursors (MALPs), in bone. Compared to other mesenchymal subpopulations, MALPs expressed Csf1 at a much higher level and this expression was further increased during aging. To investigate its role, we constructed MALP-deficient Csf1 CKO mice using Adipoq^Cre. These mice had increased femoral trabecular bone mass, but their cortical bone appeared normal. In comparison, depletion of Csf1 in the entire mesenchymal lineage using Prrx1^Cre led to a more striking high bone mass phenotype, suggesting that additional mesenchymal subpopulations secrete Csf1. TRAP staining revealed diminished osteoclasts in the femoral secondary spongiosa region of Csf1 CKO^Adipoq mice, but not at the chondral-osseous junction nor at the endosteal surface of cortical bone. Moreover, Csf1 CKO^Adipoq mice were resistant to LPS-induced calvarial osteolysis. Bone marrow cellularity, hematopoietic progenitors, and macrophages were also reduced in these mice. Taken together, our studies demonstrate that MALPs synthesize Csf1 to control bone remodeling and hematopoiesis.

## Editor's evaluation

This fundamental work advances our understanding of bone marrow adipogenic lineage precursors as a major source of Csf1 in bone and important regulator of bone remodeling. The evidence supporting the conclusion is compelling, using Adipoq-Cre-based conditional deletion of Csf1 and scRNA-seq approaches. This paper is of potential interest to scientists who study bone marrow stem/progenitor cells, bone remodeling, and metabolism.

## Introduction

Bone is maintained by a fine balance between bone formation by osteoblasts and bone resorption by osteoclasts. These two types of functional cells originate from different stem cell lineages, with the former derived from bone marrow mesenchymal stem cells (MSCs) and the latter from hematopoietic stem cells (HSCs). In addition to osteoblasts and osteoclasts, MSCs and HSCs also give rise to many other cell subpopulations that co-exist inside bone (*Bianco et al., 2010*; *Zhang et al., 2019*). Crosstalk between mesenchymal and hematopoietic subpopulations plays a critical role in bone homeostasis, and disruption of it shifts the balance of bone remodeling, leading to bone disorders (*Raggatt and Partridge, 2010*).

As a specific type of macrophage, osteoclasts are differentiated from monocytes in the myeloid subpopulation of hematopoietic lineage cells (*Boyle et al., 2003*). Colony stimulating factor 1 (Csf1), also known as macrophage colony-stimulating factor (M-Csf), is essential for osteoclastogenesis due to its actions in promoting proliferation, survival, and differentiation of monocytes and macrophages (*Stanley and Chitu, 2014*). The expression of its receptor, *Csf1r*, is low in immature myeloid precursor cells and increases as the myeloid cells mature (*Tagoh et al., 2002*). Receptor activator of NF-κB ligand (RANKL) is another indispensable cytokine for promoting osteoclastogenesis (*Kong et al., 1999*; *Suda et al., 1999*; *Kim et al., 2000*). Csf1/Csf1r signaling induces the expression of the receptor activator of NF-κB (RANK), a receptor for RANKL, to further facilitate the formation of mature osteoclasts (*Arai et al., 1999*). Through alternative splicing and proteolysis, Csf1 can be expressed into a secreted glycoprotein, a secreted proteoglycan, or a membrane-spanning cell surface glycoprotein (*Stanley et al., 2009*). A spontaneous inactivating mutation in the mouse *Csf1* coding region causes a complete loss of Csf1 protein (*Yoshida et al., 1990*). The resultant homozygous *Csf1op/op* mice are osteopetrotic due to osteoclast deficiency, but this bone abnormality recovers over the first few months of life (*Begg et al., 1993*). Similarly, a spontaneous recessive mutation (*toothless, tl*) in the rat *Csf1* gene leads to a more complete loss of osteoclasts, resulting in more severe osteopetrosis than *Csf1op/op* mice with no improvement when rats age (*Van Wesenbeeck et al., 2002*). Knocking out *Csf1r* gene in mice generates similar or even more severe skeletal phenotypes than *Csf1* deficient mice (*Dai et al., 2002*), further demonstrating the important role of Csf1/Csf1r signaling in controlling bone resorption.

Past studies have demonstrated the expression of *Csf1* in a wide range of cells, such as fibroblasts, endothelial cells, keratinocytes, astrocytes, myoblasts, breast and uterine epithelial cells (*Sehgal et al., 2021*). Primary cell culture experiments indicated that osteoblasts synthesize both soluble and membrane-bound forms of Csf1 (*Felix et al., 1996*). Interestingly, targeting expression of soluble Csf1 by osteoblast-specific promoter (osteocalcin) rescues the osteopetrotic bone defect in *Csf1op/op* mice (*Abboud et al., 2002*). Csf1 is also expressed in osteocytes (*Werner et al., 2020*). Mice with osteocyte-specific *Csf1* deficiency were generated using *Dmp1Cre*. These mice showed increased trabecular bone mass in tibiae and vertebrates (*Werner et al., 2020*). Therefore, the prevalent view is that bone forming cells, including osteoblasts and osteocytes, are the major producers of Csf1 to control osteoclast formation.

In addition to bone forming cells, bone marrow MSCs also give rise to marrow adipocytes. With the advance of single-cell transcriptomics technology, we and others recently dissected the heterogeneity of bone marrow mesenchymal lineage cells and delineated the bi-lineage differentiation trajectories of MSCs (*Baryawno et al., 2019*; *Zhong et al., 2020*; *Matsushita et al., 2020*; *Tikhonova et al., 2019*). Specifically, we identified a novel mesenchymal subpopulation termed 'marrow adipogenic lineage precursors' (MALPs) that expresses many adipocyte markers but contains no lipid droplets (*Zhong et al., 2021*). As precursors for lipid-laden adipocytes (LiLAs), MALPs exist abundantly as pericytes and stromal cells, forming an extensive 3D network inside the marrow cavity. Interestingly,

computational analysis revealed MALPs to be the most interactive mesenchymal cells with monocyte-macrophage lineage cells due to their high and specific expression of several osteoclast regulatory factors, including RANKL and Csf1 (*Yu et al., 2021*). Studies from our group and others have demonstrated that MALP-derived RANKL is critical for promoting bone resorption during physiological and pathological conditions (*Yu et al., 2021*; *Hu et al., 2021*). Since Csf1 is also important for osteoclastogenesis, here we generated *Csf1* conditional knockout mice using *Adipoq^Cre* driver and examine the role of MALP-derived Csf1 in bone homeostasis and diseases. Our findings identified MALPs as the major source of Csf1 in bone and reinforce our discovery that MALPs are a novel key player in controlling bone resorption. Furthermore, we found that MALP-derived Csf1 contributes to hematopoiesis in the bone marrow, particularly the production of macrophages.

## Results

### ScRNA-seq data suggest MALPs as the major source of Csf1 in bone

To examine the cellular sources of Csf1, we integrated scRNA-seq datasets of bone marrow cells we previously generated from 1, 3, and 16-month-old mice (*Figure 1A*; *Zhong et al., 2020*). These experiments were initially designed to analyze mesenchymal lineage cells in bone. Since they also included hematopoietic, endothelial, and mural cells, those scRNA-seq datasets indeed represent the overall cellular composition of bone tissue. Interestingly, we found that the major Csf1 producing cells are MALPs, followed by lineage committed progenitors (LCPs) and early mesenchymal progenitors (EMPs, *Figure 1B*). Aging not only increased *Csf1* expression in these cells, but also initiated its expression in some other subpopulations, such as mesenchymal cells (EMPs, late mesenchymal progenitors [LMPs], and osteoblasts), hematopoietic cells (macrophages, granulocyte progenitors, and red blood cells), endothelial cells, and mural cells. In aged mice, MALPs were markedly expanded while EMPs and LMPs shrunk in numbers (*Figure 1C*). Thus, our scRNA-seq data predict an increased amount of Csf1 in mouse bone marrow during aging. Concomitantly, *Csf1r* was mainly expressed in monocytic subpopulations, such as monocytes, macrophages, and osteoclasts, in an age-independent manner (*Figure 1B*). In line with previous reports (*Grabert et al., 2020*; *Mossadegh-Keller et al., 2013*), HSCs also expressed *Csf1r* at a low level. To validate these sequencing data, we sorted Td+ cells from the bone marrow of 3-month-old *Adipoq^Cre:Rosa26^LSL-tdTomato* (Adipoq:Td) mice, a MALP reporter line. While Td+ cells only make up ~0.5% of bone marrow cells (*Zhong et al., 2020*), qRT-PCR analysis revealed that their expression of *Csf1* is 10.2-fold higher than Td- cells (*Figure 1D*). Furthermore, bone marrow cells from 10-month-old mice expressed 5.3-fold more *Csf1* than those from 1-month-old mice (*Figure 1E*).

MALPs can further differentiate into LiLAs (*Zhong et al., 2020*). To understand whether LiLAs express *Csf1*, we analyzed a single-nucleus RNA-sequencing (snRNA-seq) dataset from white adipose tissue of 16-week-old mice that contains LiLAs (GSE number 176171, *Figure 1F*; *Emont et al., 2022*). Surprisingly, LiLAs did not express *Csf1* at a detectable level. Instead, adipose stem and progenitor cells (ASPCs) and mesothelial cells were the major Csf1-producing cells, albeit their expression levels were much lower compared to MALPs in the bone marrow (*Figure 1G*). As expected, the highest *Csf1r* expression was detected in adipose tissue resident macrophages (*Figure 1G*). Furthermore, a companied article published together with this one used qRT-PCR and immunostaining approaches to clearly show that in the bone marrow, Csf1 is expressed in MALPs but not in LiLAs (*Inoue et al., 2023*). In addition, we also detected strong *Csf1* expression in Adipoq+ cells from the control bone marrow (i.e. non-pathologic) acquired from a recent scRNA-seq study examining human multiple myeloma microenvironment (*de Jong et al., 2021*). Taken together, our scRNA-seq data indicate that MALPs are the major Csf1-producing cells in bone tissue.

### Mice with Csf1 deficiency in MALPs have high trabecular bone mass

Our previous studies demonstrate that adipocyte-specific *Adipoq^Cre* targets MALPs and LiLAs, but not osteoblasts and osteocytes, in young and adult mice (*Zhong et al., 2020*; *Yu et al., 2021*). Thus, we constructed *Adipoq^Cre Csf1^flox/flox* (*Csf1* CKO^Adipoq) mice to investigate the role of MALP-derived Csf1. These mice were born at the expected frequency. They grew normally with comparable body weight, femoral length, and tooth eruption as *Csf1^flox/flox* and *Csf1^flox/+* siblings, which are considered controls in this study (*Figure 2A–C*). Their peripheral white adipose tissue also appeared normal, with similar

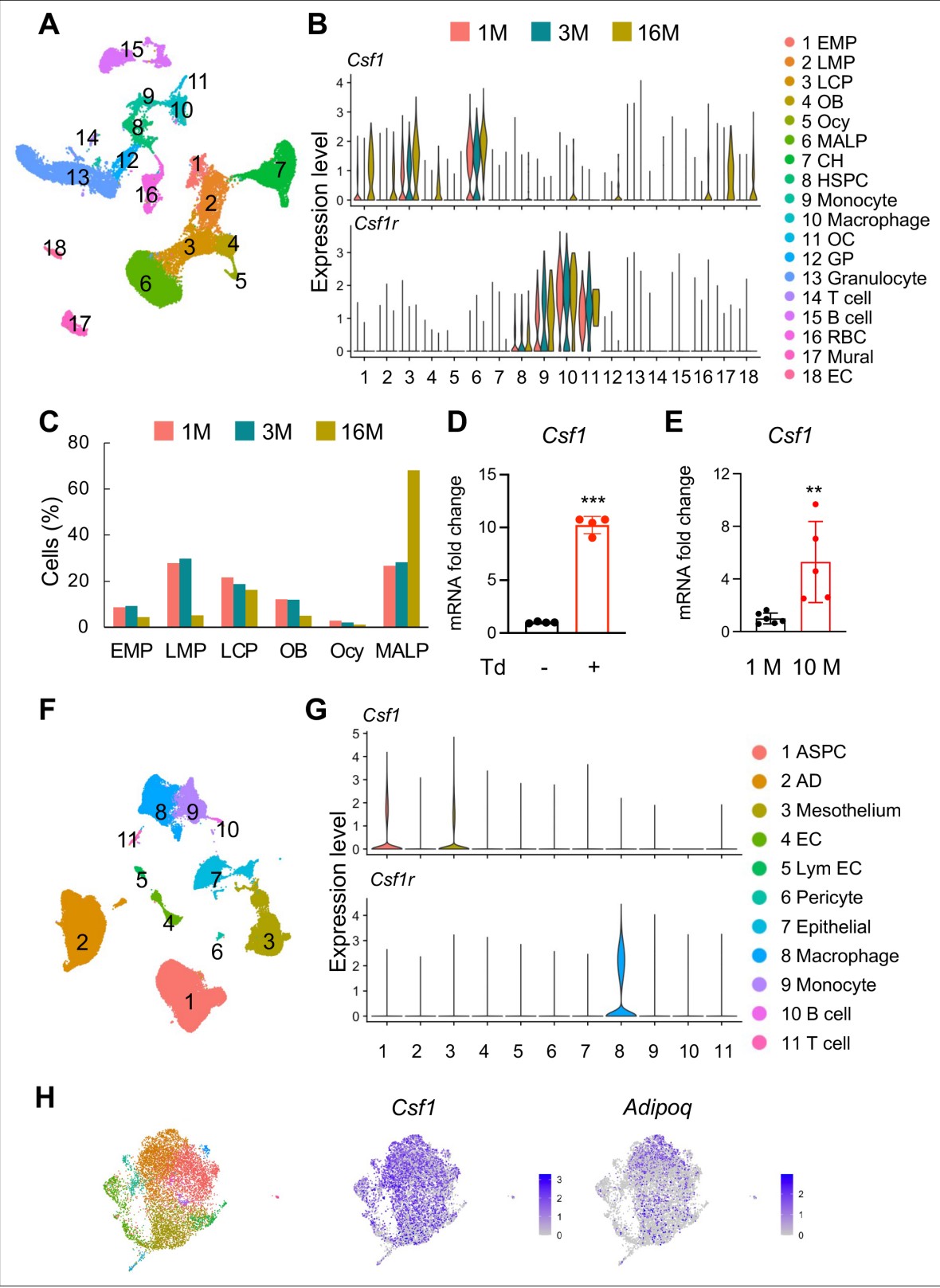

**Figure 1.** *Csf1* expression in bone is mainly contributed by MALPs in an age-dependent manner. (**A**) The integrated scRNA-seq dataset of sorted bone marrow Td+ cells from 1, 3, and 16-month-old *Col2a1^Cre^:Rosa26^LSL-tdTomato^* (Col2:Td) mice mice (n=11 mice). The UMAP plot is presented to show cell clustering. (**B**) Violin plots of *Csf1* and its receptor *Csf1r* in bone marrow cells at different ages. EMP: early mesenchymal progenitor; LMP: late mesenchymal progenitor; LCP: lineage committed progenitor; OB: osteoblast; Ocy: osteocyte; CH: chondrocyte; EC: endothelial cell; HSPC:

*Figure 1 continued on next page*

*Figure 1 continued*

hematopoietic stem and progenitor cell; OC: osteoclast; GP: granulocyte progenitor; RBC: red blood cell; EC: endothelial cell. (**C**) The percentages of bone marrow mesenchymal subpopulations are quantified based on UMAP distribution. (**D**) qRT-PCR analysis of *Csf1* expression in sorted Td+ and Td- bone marrow cells from Adipoq:Td mice mice at 3 months of age. ***, p<0.001 Td+ vs Td- cells. (**E**) qRT-PCR analysis of *Csf1* expression in bone marrow from young (1 month of age) and aged (10 months of age) control mice. **, p<0.01 10 M vs 1 M. (**F**) SnRNA-seq analysis of inguinal and perigonadal adipose tissues from 16-week-old mice. The UMAP plot is presented to show cell clustering. (**G**) Violin plots of Csf1 and its receptor *Csf1r* in individual cell subpopulation from peripheral adipose tissue. ASPC: adipose stem and progenitor cell; AD: adipocyte; EC: endothelial cell; Lym EC: lymphatic endothelial cell. (**H**) ScRNA-seq analysis of mesenchymal cells from human control bone marrow. These bone marrow samples were obtained by either sternal aspiration from donors undergoing cardiothoracic surgery or by manual bone marrow collection from femur heads collected after hip replacement surgery. The UMAP plots are presented to show cell clustering on the left and Csf1 and Adipoq expression on the right.

The online version of this article includes the following source data for figure 1:

**Source data 1.** Full dataset for *Figure 1D*.

**Source data 2.** Full dataset for *Figure 1E*.

adipocyte size, number, and vasculature as control mice (*Figure 2—figure supplement 1*). The *Csf1* mRNA level in CKO bone marrow was reduced to 43% of control bone marrow (*Figure 2D*). This result not only validates the knockout efficiency but also supports MALPs as the major source of Csf1 in the bone marrow. On the contrary, Csf1 expression in the cortical bone was not changed (*Figure 2D*), suggesting that osteocytes are not affected in *Csf1* CKO*Adipoq* mice. Binding to the same receptor Csf1R, Il34 shares a redundant role with Csf1 (*Lelios et al., 2020*). However, ELISA analysis found no change in IL-34 amount in the CKO bone marrow (control 1.27±0.44 µg/ml vs CKO 1.08±0.55 µg/ml, n=5–6 mice), suggesting no compensatory increase of Il34 in CKO.

MicroCT analysis of femoral trabecular bone revealed substantial osteopetrosis in these CKO mice starting from 3 months of age (*Figure 2E and F*). Compared to control siblings, CKO mice showed significantly elevated bone volume fraction (BV/TV, 1.7-fold) and bone mineral density (BMD, 1.4-fold) at 3 months of age and additional accrual (3.5- and 2.0-fold, respectively) at 6 months of age. These changes were accompanied by increased trabecular number (Tb.N, 14% and 31% at 3 and 6 months of age, respectively) and trabecular thickness (Tb.Th, 27% at 6 months of age), and decreased trabecular separation (Tb.Sp, 13% and 26% at 3 and 6 months of age, respectively). The reduced structure model index (SMI) confirmed the improved trabecular bone microarchitecture in CKO mice. Meanwhile, femoral cortical bone structure, tissue mineral density (TMD), and mechanical properties were not altered (*Figure 2—figure supplement 2*). To our surprise, while MALPs are present in vertebral bone marrow (*Zhong et al., 2020*), we did not detect any structural changes in vertebral trabecular bone (*Figure 2—figure supplement 3*).

## Csf1 from mesenchymal lineage cells other than MALPs regulate bone structure

To explore whether Csf1 from MALPs plays a dominant role in regulating bone structure, we generated *Prrx1Cre Csf1flox/flox* (*Csf1* CKO*Prrx1*) mice to knockout *Csf1* in all mesenchymal lineage cells in bone (*Logan et al., 2002*), including MALPs. Like *Csf1* CKO*Adipoq* mice, these *Csf1* CKO*Prrx1* mice had normal tooth eruption (data not shown). At 2–3 months of age, these mice showed more drastic skeletal abnormalities than *Csf1* CKO*Adipoq*. Their femoral length was reduced by 13% (*Figure 3A*), indicating a defect in growth plate development. Their trabecular BV/TV and BMD were remarkably increased by 10.3-fold and 5.7-fold, respectively, due to increased Tb.N (2.3-fold) and Tb.Th (2.7-fold), and decreased Tb.Sp (49%, *Figure 3B–D*). Furthermore, their cortical bone was also enlarged. We observed significantly increased periosteal perimeter (Ps.Pm), endosteal perimeter (Ec.Pm), and cortical bone area (Ct.Ar) in *Csf1* CKO*Prrx1* mice compared to control mice (*Figure 3E and F*). Interestingly, the endosteal surface in *Csf1* CKO*Prrx1* mice was not as smooth as that in control mice, due to extensive trabecular bone remnants at the midshaft area. Since *Prrx1Cre* labels the entire mesenchymal lineage cells in bone, including chondrocytes, mesenchymal progenitors, osteoblasts, and osteocytes, these data suggest that mesenchymal cells other than MALPs also produce Csf1 to regulate bone structure.

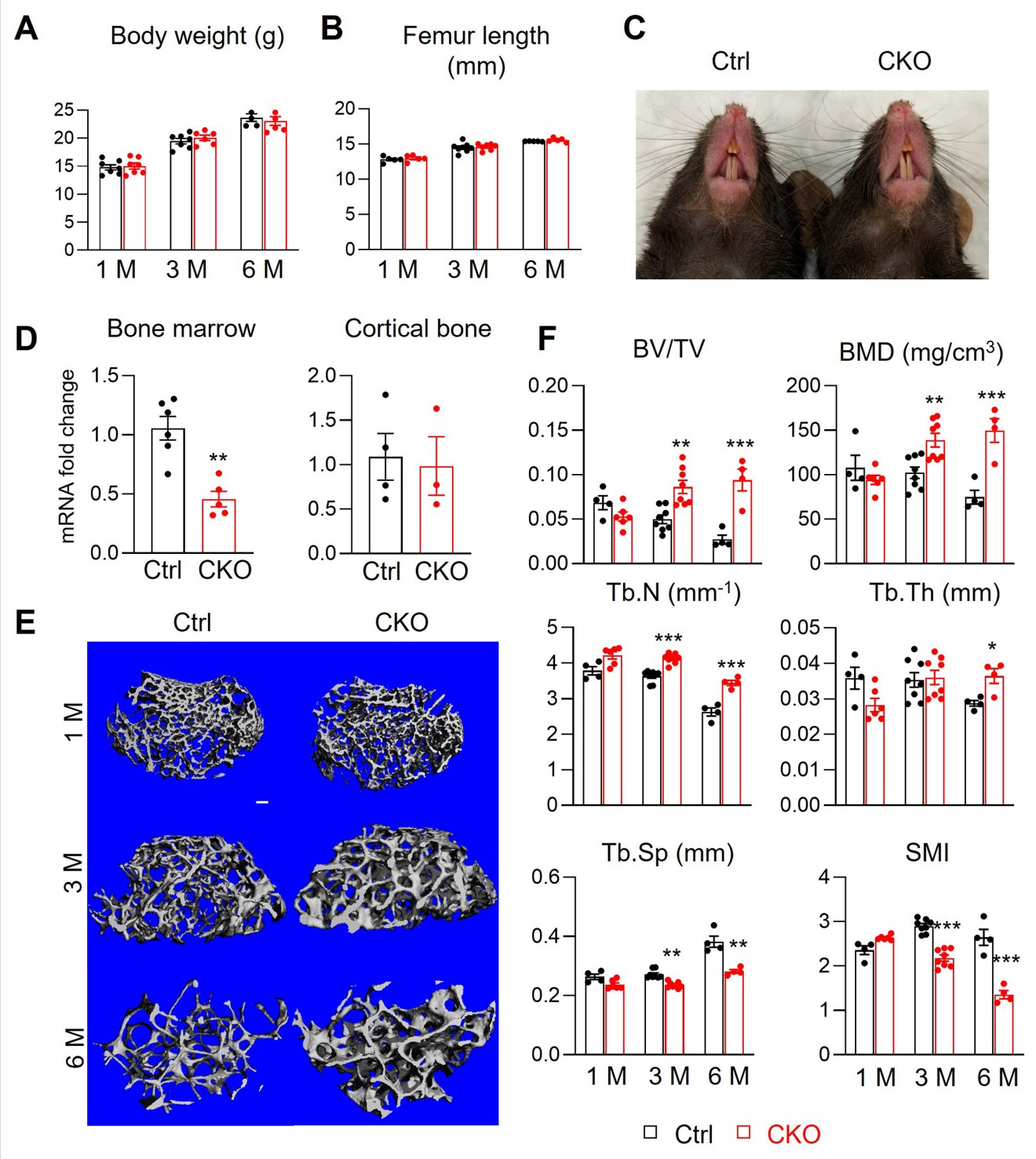

**Figure 2.** *Csf1* CKO*Adipoq* mice have high trabecular bone mass in long bone. (**A, B**) *Csf1* CKO*Adipoq* mice have normal body weight (**A**) and femoral length (**B**) at 1, 3, and 6 months of age. n=5–7 mice/group. (**C**) Tooth eruption is also normal in CKO mice at 1 month of age. (**D**) qRT-PCR analysis of *Csf1* mRNA in bone marrow and cortical bone of control (Ctrl) and *Csf1* CKO*Adipoq* mice at 3 months of age. n=3–6 mice/group. (**E**) 3D microCT reconstruction of femoral secondary spongiosa region from 1-, 3-, and 6-month-old mice reveals a drastic increase of trabecular bone in female *Csf1* CKO*Adipoq* mice compared to control mice. Scale bar = 100 µm. (**F**) MicroCT measurement of trabecular bone structural parameters. BV/TV: bone volume fraction; BMD: bone mineral density; Tb.N: trabecular number; Tb.Th: trabecular thickness; Tb.Sp: trabecular separation; SMI: structural model index. n=4–8 mice/group. *, p<0.05; **, p,0.01; ***, p<0.001 CKO vs control.

The online version of this article includes the following source data and figure supplement(s) for figure 2:

*Figure 2 continued on next page*

*Figure 2 continued*

**Source data 1.** Full dataset for *Figure 2A*.

**Source data 2.** Full dataset for *Figure 2B*.

**Source data 3.** Full dataset for *Figure 2D*.

**Source data 4.** Full dataset for *Figure 2F*.

**Figure supplement 1.** *Csf1* CKO*^{Adipoq}* mice have normal subcutaneous fat pad.

**Figure supplement 2.** *Csf1* CKO*^{Adipoq}* mice have normal cortical bone structure and mechanical properties.

**Figure supplement 2—source data 1.** Full dataset for *Figure 2—figure supplement 2B*.

**Figure supplement 2—source data 2.** Full dataset for *Figure 2—figure supplement 2C*.

**Figure supplement 3.** *Csf1* deficiency in MALPs does not affects vertebral bone.

**Figure supplement 3—source data 1.** Full dataset for *Figure 2—figure supplement 3B*.

## Csf1 deficiency in MALPs suppresses bone resorption but does not affect bone formation

To delineate the cellular mechanism of high bone mass in *Csf1* CKO*^{Adipoq}* mice, we quantified osteoclasts at different locations of long bones (*Figure 4A and B*). TRAP+ osteoclasts in the secondary spongiosa (SS) were remarkably reduced by 58%. However, no changes in osteoclasts were detected at the chondral-osseous junction (COJ) and the endosteal surface of cortical bone. To examine osteoclast progenitors in *Csf1* CKO*^{Adipoq}* mice, we harvested bone marrow macrophages (BMMs) for in vitro osteoclastogenesis assay. Cells from both control and CKO mice generated similar numbers of osteoclasts (*Figure 4C and D*), suggesting that the differentiation ability of osteoclast progenitors is maintained in *Csf1* CKO*^{Adipoq}* mice.

We also measured bone formation activity in 3-month-old *Csf1* CKO*^{Adipoq}* femurs. No significant difference was detected in Osterix+ osteoblasts at the trabecular bone surface in control and CKO femurs (*Figure 4E and F*). Dynamic bone histomorphometry showed a similar distance between two fluorescent dyes injected at 9 and 2 days, respectively, in control and CKO mice before euthanization (*Figure 4G*). Quantification revealed comparable mineral apposition rate (MAR), mineralization surface (MS/BS), and bone formation rate (BFR/BS, *Figure 4H*). Furthermore, bone resorption marker (CTX-1) was significantly reduced in CKO serum, while bone formation marker (PINP) was not changed (*Figure 4I*).

## Pathological bone loss is attenuated in Csf1 CKO*^{Adipoq}* mice

Lipopolysaccharide (LPS) injection above mouse calvaria induces osteolysis that mimics bacteria-induced bone loss. Since we previously identified the existence of MALPs in calvarial bone marrow (*Yu et al., 2021*), we applied this osteolysis model to *Csf1* CKO*^{Adipoq}* mice. One week after LPS injection, we found a drastic increase in bone destruction in control calvaria, but very minor destruction in *Csf1* CKO*^{Adipoq}* calvaria (*Figure 5A and B*). H&E staining showed that calvarial bone marrow is surrounded by a thin layer of cortical bone (*Figure 5C*). After the LPS injection, normal calvarial structure, including suture and cortical bone, were mostly eroded, and filled with inflammatory cells in control mice, but unaltered in CKO mice. TRAP staining revealed that LPS-induction of TRAP+ osteoclasts in calvaria is mostly attenuated in CKO mice (*Figure 5C and D*). Taken together, our data showed that MALPs-derived Csf1 is important for osteoclastogenesis in diseased bones.

## Csf1 CKO*^{Adipoq}* mice have defective hematopoiesis in the bone marrow

Csf1 plays multifaceted roles in bone and blood tissues (*Sehgal et al., 2021*). We next investigated the role of MALPs-derived Csf1 in regulating hematopoiesis and bone marrow vasculature. Interestingly, bone marrow cellularity started to decline in *Csf1* CKO*^{Adipoq}* mice at 3 months of age (18%) and further reduced at 6 months (43%, *Figure 6A*), suggesting that hematopoiesis is suppressed. Macrophages are the major cell target of Csf1. Immunostaining revealed that MALPs directly contact F4/80+ macrophages (*Figure 6B*) and that there are fewer macrophages in the CKO bone marrow (*Figure 6C*). Flow cytometric analysis confirmed that Cd11b+F4/80+ bone marrow macrophages were reduced by 36%, 65%, and 51%, in CKO mice compared to control at 1, 3, and 6 months of age, respectively (*Figure 6D*).

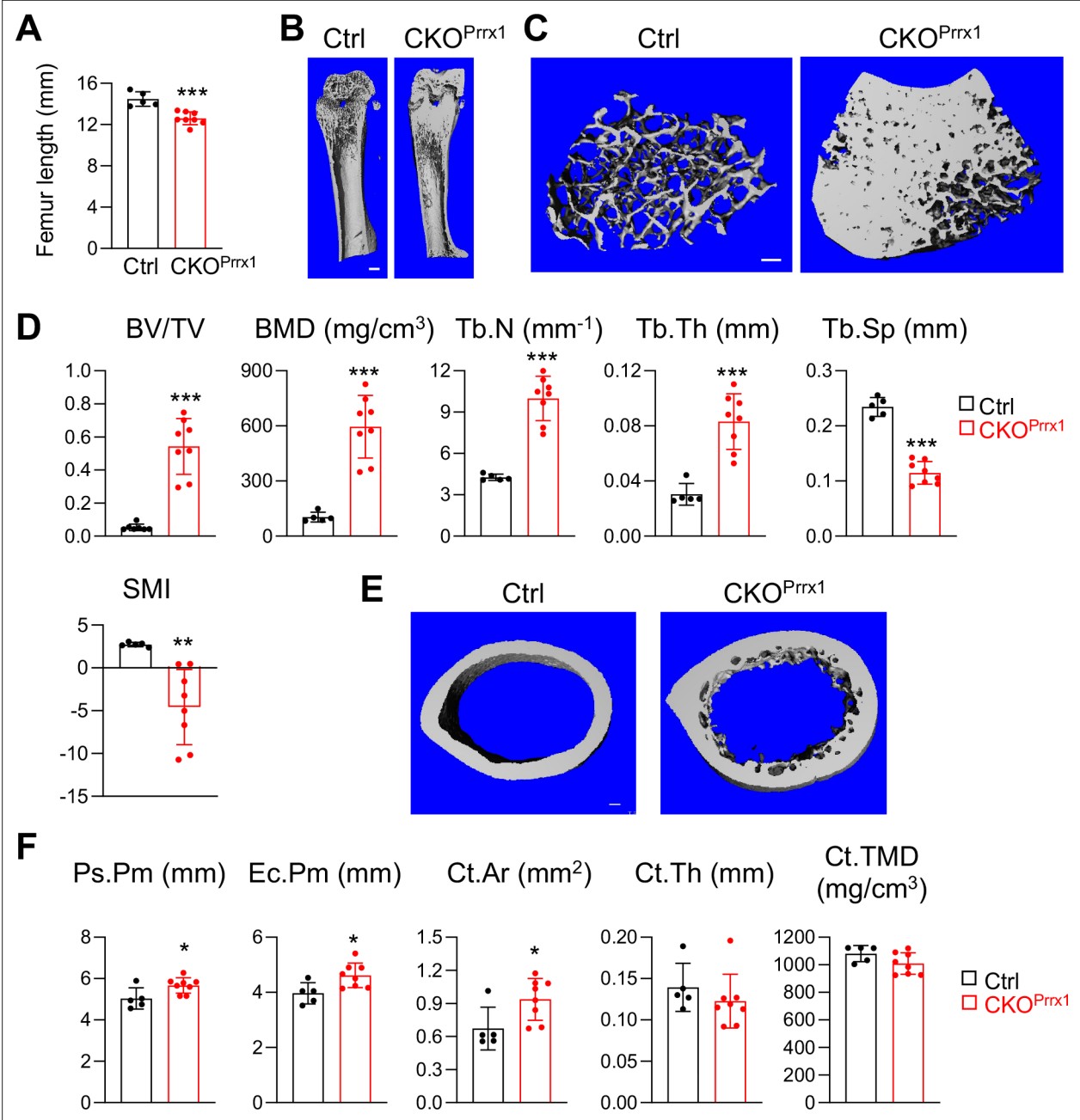

**Figure 3.** *Csf1* depletion in all mesenchymal cells using *Prrx1Cre* affects bone growth and causes severe osteopetrosis. (**A**) Femur length measurement in *Csf1* CKO*Prrx1* mice at 2–3 months of age (n=5–8 mice/group). (**B**) 3D microCT reconstruction of whole femurs from control and *Csf1* CKO*Prrx1* mice. Scale bar = 1 mm. (**C**) 3D microCT reconstruction of femoral secondary spongiosa region from control and *Csf1* CKO*Prrx1* mice. Scale bar = 100 μm. (**D**) MicroCT measurement of trabecular bone structural parameters. BV/TV: bone volume fraction; BMD: bone mineral density; Tb.N: trabecular number; Tb.Th: trabecular thickness; Tb.Sp: trabecular separation; SMI: structural model index. n=5–8 mice/group. (**E**) 3D microCT reconstruction of femoral cortical bone. Scale bar = 100 μm. (**F**) MicroCT measurement of cortical bone structural parameters. Ps.Pm: periosteal perimeter; Ec.Pm: endosteal perimeter; Ct.TMD: cortical tissue mineral density. Ct.Ar: cortical area; Ct.Th: cortical thickness. n=5–8 mice/group. *, $p < 0.05$; ***, $p < 0.001$ CKO*Prrx1* vs control.

The online version of this article includes the following source data for figure 3:

**Source data 1.** Full dataset for *Figure 3A*.

**Source data 2.** Full dataset for *Figure 3D*.

**Source data 3.** Full dataset for *Figure 3F*.

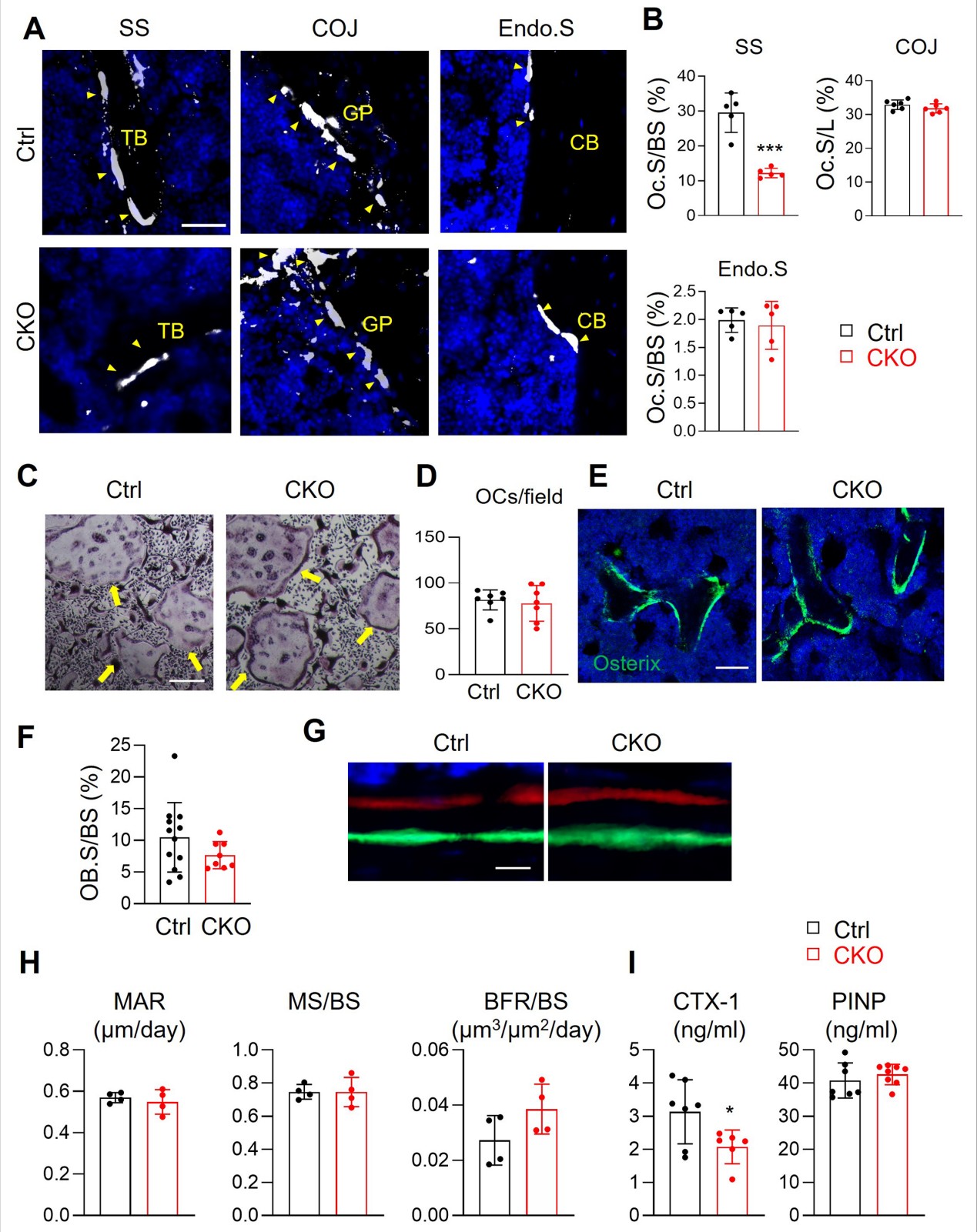

**Figure 4.** *Csf1* deletion in MALPs suppresses bone resorption but not bone formation. (**A**) Representative fluorescent TRAP staining images of femoral long bones from control and *Csf1* CKO*^Adipoq^* mice at 3 months of age show TRAP+ osteoclasts at different skeletal sites: secondary spongiosa (SS), chondro-osseous junction (COJ), and endosteal surface (Endo.S). TB: trabecular bone; CB: cortical bone. Scale bar = 50 μm. (**B**) Quantification of osteoclast surface (Oc.S) at three skeletal sites. BS: bone surface. L: COJ length. n=5 mice/group. ***, p<0.001 CKO vs control. (**C**) Representative

*Figure 4 continued on next page*

*Figure 4 continued*

TRAP staining images of osteoclast culture derived from control and *Csf1* CKO[Adipoq] BMMs at 7 days after addition of RANKL and Csf1. Arrows point to mature osteoclasts. Scale bar = 200 μm. (**D**) Quantification of TRAP+ multinucleated cells (>3 nuclei/cell) per field. n=7 mice/group. (**E**) Representative Osterix staining of trabecular bone from control and *Csf1* CKO[Adipoq] femurs. Scale bar = 50 μm. (**F**) Quantification of osteoblast surface (OB.S). BS, bone surface. n=8–12 mice/group. (**G**) Representative double labeling of trabecular bone from control and *Csf1* CKO[Adipoq] femurs. (**H**) Bone formation activity is quantified. MAR: mineral apposition rate; MS: mineralizing surface; BFR: bone formation rate. n=4 mice/group. (**I**) Serum ELISA analysis of bone resorption marker (CTX-1) and formation marker (PINP) in control and CKO mice. n=6–8 mice/group. *, p<0.05 CKO vs control.

The online version of this article includes the following source data for figure 4:

**Source data 1.** Full dataset for *Figure 4B*.

**Source data 2.** Full dataset for *Figure 4D*.

**Source data 3.** Full dataset for *Figure 4F*.

**Source data 4.** Full dataset for *Figure 4H*.

**Source data 5.** Full dataset for *Figure 4I*.

In addition to macrophages, we also found that hematopoietic stem and progenitor cells (HSPCs) are reduced in the bone marrow of *Csf1* CKO[Adipoq] mice (**Figure 6E**). While only lineage-cKit+ (LK) cells displayed a significant decrease, more primitive progenitors, such as lineage-Sca1 +cKit + (LSK) cells, multipotent progenitors (MPP), and SLAM LSK, showed a trend of reduction. Moreover, monocytes and erythroid progenitors were also decreased but other myeloid and lymphoid cells, such as neutrophils, eosinophils, B cells, and T cells, remained the same (**Figure 6F**). However, these changes did not result in significant alterations in peripheral blood compositions and spleen weight (**Figure 6—figure supplement 1**).

A previous study reported a reduction of bone marrow vasculature in the primary spongiosa area in mice with *Csf1* global deficiency (**Harris et al., 2012**). However, we did not detect obvious changes in marrow vasculature in *Csf1* CKO[Adipoq] mice (**Figure 6G and H**), indicating that Csf1 from cells other than MALPs regulates angiogenesis. In summary, MALPs-derived Csf1 is required for regulating hematopoiesis, particularly macrophage production, in the bone marrow.

## Discussion

In bone, mesenchymal stem and progenitor cells undergo two divergent differentiation routes to produce osteogenic and adipogenic cells, respectively. In the past, bone research has mostly centered on osteogenic cells, including osteoblasts and osteocytes, because of their bone matrix synthesis ability, but paid less attention to marrow adipogenic cells. Recently, reports from our group and others discovered a group of marrow mesenchymal cells that express most adipocyte markers but do not contain lipid droplets (**Zhong et al., 2020**; **Mukohira et al., 2019**; **Zou et al., 2020**; **Zhang et al., 2021a**). Those cells are termed either 'MALPs' based on their differentiation status or 'MACs' (marrow Adipoq + cells) based on their specific expression of Adipoq. Interestingly, our previous scRNA-seq study revealed that those adipogenic cells specifically and highly express two essential osteoclast regulatory factors, RANKL and Csf1 (**Zhong et al., 2020**; **Zhong et al., 2021**). In our last report, we demonstrated that RANKL from MALPs promotes osteoclast formation and bone loss under normal and diseased conditions (**Yu et al., 2021**). In this study, we found that Csf1 from MALPs shares similar action in regulating bone resorption. Since all aspects of osteoclast formation and functions are regulated by RANKL and Csf1, our data firmly establish MALPs as one of the major bone cell types that control bone resorption.

In our previous study of mice with RANKL deficiency in MALPs (*Tnfsf11* CKO[Adipoq] mice), we proposed that osteoclast formation is controlled by various mesenchymal subpopulations in a site-dependent fashion. Our current research further substantiates this conclusion. We found that *Csf1* CKO[Adipoq] mice, similar to *Tnfsf11* CKO[Adipoq] mice, exhibit trabecular bone phenotype but have normal long bone growth and cortical bone structure. However, *Csf1* CKO[Prrx1] mice showed much more severe skeletal phenotypes. These mice have shortened long bones, in line with previous reports that global Csf1 deficiency in rodents is accompanied by progressive chondrodysplasia of the growth plate and altered endochondral ossification (**Harris et al., 2012**; **Devraj et al., 2004**). These data imply chondrocyte-derived Csf1 regulates cartilage-to-bone remodeling during endochondral ossification. *Csf1* CKO[Prrx1]

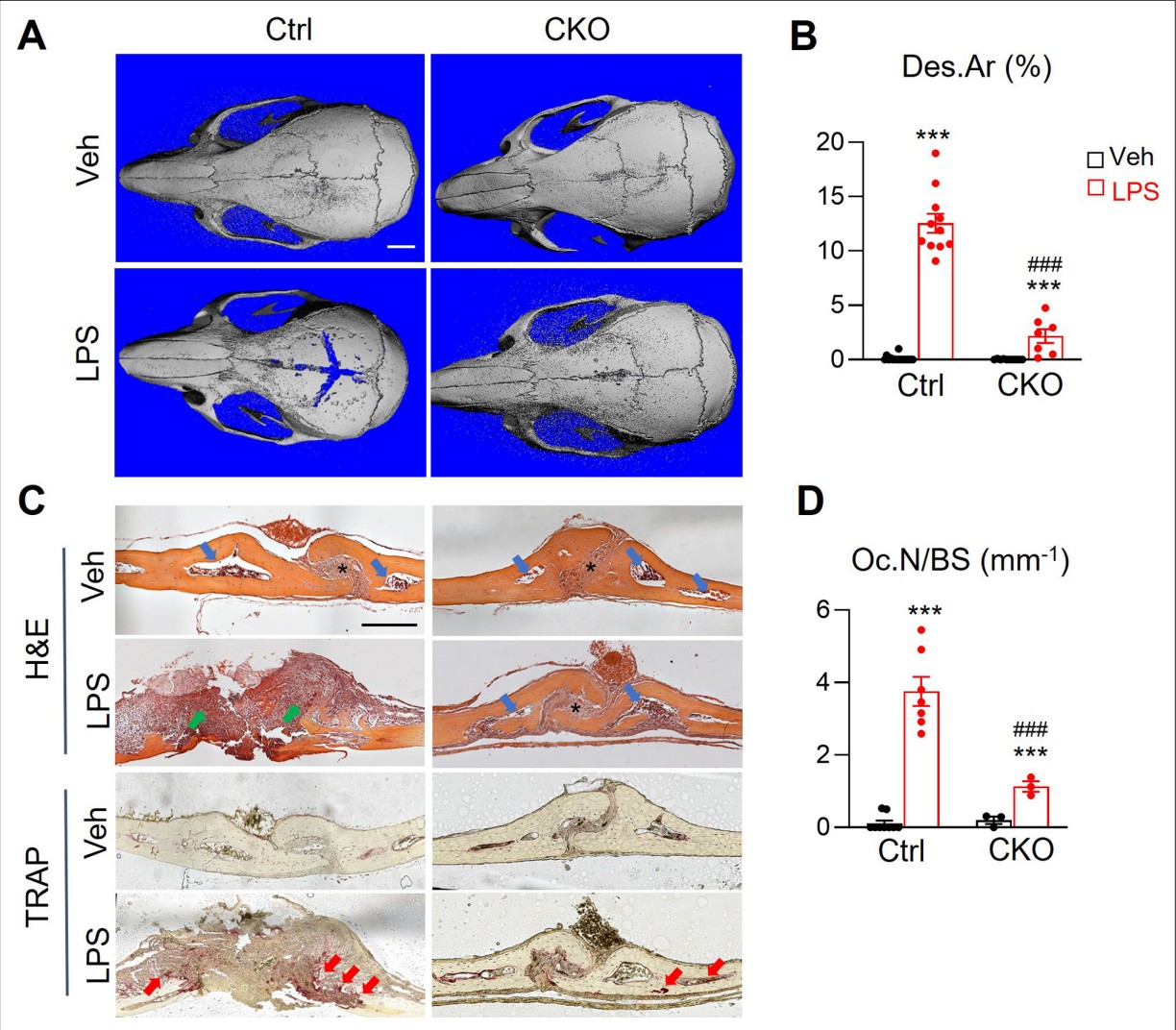

**Figure 5.** *Csf1* CKO*Adipoq* mice are protected from LPS-induced calvarial bone lesions. (**A**) Representative 3D microCT reconstruction of mouse calvaria after 1 week of vehicle (veh, PBS) or LPS injection. Scale bar = 2 mm. (**B**) The percentage of bone destruction area (Des.Ar) in calvaria was quantified. n=7–16 mice/group. (**C**) Representative images of H&E and TRAP-stained coronal sections. In the H&E-stained images, * indicates suture position and blue arrows point to calvarial bone marrow region. Green arrows are inflammatory cells in the eroded calvariae after LPS injection in control mice. In the TRAP-stained images, red arrows point to TRAP+ osteoclasts. Scale bar = 200 μm. (**D**) Quantification of osteoclast number (Oc.N) in calvaria. n=3–9 mice/group. ***, p<0.001 LPS vs Veh; ###, p<0.001 CKO vs control.

The online version of this article includes the following source data for figure 5:

**Source data 1.** Full dataset for *Figure 5B*.

**Source data 2.** Full dataset for *Figure 5D*.

mice also have thickened cortical bones, suggesting a role of osteocytes in regulating cortical bone structure. Note that scRNA-seq data generated from our group and others do not contain hypertrophic chondrocytes and mature osteocytes, likely due to the difficulties of collecting these matrix-embedding cells during tissue digestion and cell sorting. Thus, we cannot use the sequencing data to compare the Csf1 expression level between those cells and MALPs.

To our surprise, while high trabecular bone mass was evident in long bones, no changes were detected in the vertebrae of *Csf1* CKO*Adipoq* mice. Based on our scRNA-seq data, *Adipoq* expression is highly specific for MALPs. Since *Csf1* is also expressed at a considerable level in LCPs, we believe that LCP-derived Csf1, which should not be affected in *Csf1* CKO*Adipoq* mice, might be sufficient for promoting osteoclastogenesis in vertebrae. On the contrary, while *Rankl* shares a similar expression pattern as *Csf1* in the scRNA-seq data, *Tnfsf11* CKO*Adipoq* mice show a strong high trabecular bone

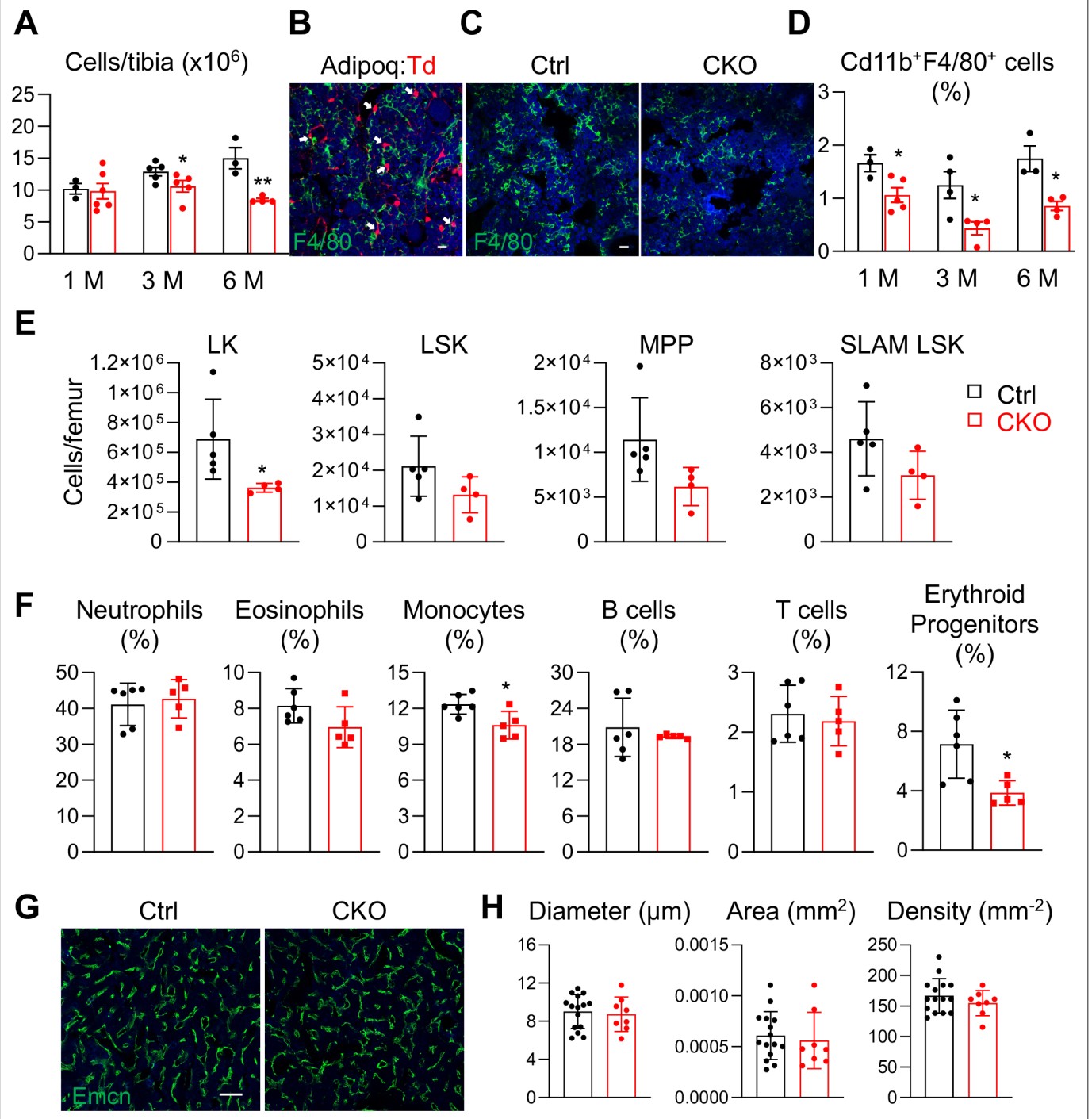

**Figure 6.** Bone marrow cellularity, hematopoietic progenitors, and macrophages are reduced in *Csf1* CKO*Adipoq* mice. (**A**) Bone marrow cellularity was quantified in control and *Csf1* CKO*Adipoq* mice at 1, 3, and 6 months of age. *, p<0.05; **, p<0.01 CKO vs control. (**B**) Fluorescent staining of F4/80 was performed on the long bones of Adipoq:Td mice. White arrows point to Adipoq+ cells (MALPs) directly contact with F4/80+ macrophages in the bone marrow. Scale bar = 20 μm. (**C**) Representative fluorescent images of F4/80 staining in control and *Csf1* CKO*Adipoq* bone marrow. Scale bar = 20 μm. (**D**) Flow analysis of Cd11b+F4/80+bone marrow macrophages at 1, 3, and 6 months of age. *, p<0.05 CKO vs control. (**E**) Cell counts of hematopoietic stem and progenitor cells. LK = Lineage-cKit+, LSK = Lineage-Sca1+cKit+, MPP = Lineage-Sca1+cKit+CD48+CD150-, SLAM LSK = Lineage-Sca1+cKit+CD48-CD150+. n=4–5 mice/group. *, *P*<0.05 CKO vs control. (**F**) Flow analysis of bone marrow hematopoietic subpopulations. Neutrophil = CD45+CD11b+Ly6G+; estrophil = CD45+CD11b+Ly6 G-CD170+; monocyte = CD45+CD11b+Ly6G-CD170-Ly6C+; B cell = CD45+Ter119 CD3-CD45R/B220+; T cell = CD45+Ter119 CD3+; erythroid progenitors = CD45+Ter119+. n=5–6 mice/group. *, p<0.05 CKO vs control. (**G**) Representative

*Figure 6 continued on next page*

*Figure 6 continued*

fluorescent images of bone marrow vasculature stained by Endomucin (Emcn). Scale bar = 100 μm. (**H**) Quantification of bone marrow vessel diameter, density, and area. n=8–15 mice/group.

The online version of this article includes the following source data and figure supplement(s) for figure 6:

**Source data 1.** Full dataset for *Figure 6A*.

**Source data 2.** Full dataset for *Figure 6D*.

**Source data 3.** Full dataset for *Figure 6E*.

**Source data 4.** Full dataset for *Figure 6F*.

**Source data 5.** Full dataset for *Figure 6H*.

**Figure supplement 1.** Peripheral blood and spleen appear normal in 3-month-old *Csf1* CKO$^{Adipoq}$ mice.

**Figure supplement 1—source data 1.** Full dataset for *Figure 6—figure supplement 1A*.

**Figure supplement 1—source data 2.** Full dataset for *Figure 6—figure supplement 1B*.

mass phenotype in the vertebrae (*Yu et al., 2021*). Future studies identifying specific markers for LCPs and designing LCP-specific *Csf1* knockout mice could add another missing piece of mesenchymal subpopulation that contributes to bone resorption and hematopoiesis.

It is traditionally thought that osteogenic cells, particularly osteocytes, are the major source of RANKL and Csf1 (*Kitaura et al., 2020*). Thus, bone forming cells and bone resorbing cells constitute a feedback loop to maintain the fine balance of bone remodeling. Our findings add MALPs as a new cell type governing this balance. First, depletion of RANKL or Csf1 in MALPs results in a similar or even more severe osteopetrotic phenotype than depletion of those factors in osteoblasts/osteocytes. In this study, we found that femoral trabecular bone mass increased in *Csf1* CKO$^{Adipoq}$ mice by ~73% at 3 months of age, comparable to the trabecular bone mass increase in *Csf1* CKO$^{Dmp1}$ mice at 4 months of age (*Werner et al., 2020*). In our previous study, we found that *Tnfsf11* CKO$^{Adipoq}$ mice develop osteopetrosis at 1 month of age while *Tnfsf11* CKO$^{Dmp1}$ mice at the same age have normal bone mass (*Yu et al., 2021*). Second, MALPs also suppress bone formation. Using a cell ablation model, we found that depletion of MALPs results in de novo bone formation in diaphyseal bone marrow, which cannot be rescued by transplant of WT white adipose tissue (*Zhong et al., 2020*). Studies by Zuo et al. validated this phenotype and further identified that MALPs highly express two potent BMP signaling inhibitors, chordin-like1 (Chrdl1) and gremlin1 (Grem1), to suppress osteogenic differentiation of bone marrow MSCs (*Zou et al., 2020*). By acting on both bone resorption and formation, we believe that MALPs limit the overall trabecular bone mass to provide enough marrow space for blood production. This is also consistent with our recent findings that MALPs highly express hematopoietic factors and that MALPs mediate bone marrow blood cell recovery after radiation damage (*Zhong et al., 2022*).

Unlike RANKL, which is highly specific for mature osteoclast formation, Csf1 has much broader actions in tissue development, homeostasis, and repair (*Sehgal et al., 2021*). Our previous study with *Tnfsf11* CKO$^{Adipoq}$ mice mainly detected bone phenotypes due to suppressed osteoclast formation. We did not notice obvious changes in bone marrow cellularity in those mice. However, *Csf1* CKO$^{Adipoq}$ mice have hematopoietic phenotypes. Compared to control mice, their bone marrow cellularity, macrophages, and hematopoietic progenitors are reduced. In line with the well-known fact that Csf1 drives myelopoiesis from HSCs (*Mossadegh-Keller et al., 2013*), these results further indicate that MALP-derived Csf1 plays an important role in hematopoiesis, particularly the production of marrow-resident macrophages. In contrast, a recent study using *Lepr*$^{Cre}$ to knockout *Csf1* did not find any changes in mouse bone marrow cellularity, marrow blood cells (including HSCs and macrophages), and peripheral blood composition (*Zhang et al., 2021b*). Since MALPs also express *Lepr* (*Zhong et al., 2021*), we acknowledge the inconsistency between those data and ours. Future experiments should be performed to examine whether *Csf1* CKO$^{Lepr}$ mice have suppressed bone resorption and to compare the *Csf1* expression pattern in bones of CKO$^{Lepr}$ and CKO$^{Adipoq}$ mice.

The Csf-1 isoforms are synthesized from a full-length and a truncated precursor (*Pixley and Stanley, 2004*). The secreted isoforms are derived by differential proteolysis from the full-length and the cell-surface isoform is derived from the truncated precursor encoded by an alternatively spliced mRNA in which the regions encoding the proteolytic cleavage sites are spliced out. The N-terminal 150 amino acids of both precursors are identical and sufficient for in vitro biological activity. In the companied

article published together with this one, immunostaining shows Csf1 staining in MALPs, suggesting that MALPs do express the membrane form of Csf1. Whether MALPs express secreted form of Csf1 is still unclear. A previous study found that the scant bone marrow macrophages in *op/op* mice are partially restored by secreted Csf1 injections (*Cecchini et al., 1994*), indicating that bone marrow macrophages could be regulated by both secreted and membrane-bound Csf1, possibly originated from MALPs.

In summary, the present analysis of *Csf1* CKO$^{Adipoq}$ mice contributes significantly to our understanding of cellular sources of Csf1 that regulate bone remodeling and myeloid blood cell production. Together with our previous analysis of *Tnfsf11* CKO$^{Adipoq}$ mice, our research demonstrates adipogenic progenitors in the bone marrow are an important mesenchymal subpopulation that controls osteoclastogenesis. The general functions of Csf1 on monocyte lineage cells implicate that MALPs might have broad roles in inflammation and tissue regeneration. Therefore, future directions should be aimed to develop novel therapeutic strategies for bone and blood-related disorders by specifically targeting MALPs.

## Methods

### Analysis of scRNA-seq datasets

Pre-aligned scRNA-seq matrix files were acquired from GEO GSE145477 and GSE176171 (mouse) and EMBL-EBI E-MTAB-9139 (human). Standard Seurat pipeline (*Stuart et al., 2019*) was used for filtering, normalization, variable gene selection, dimensionality reduction analysis and clustering. For the integrated dataset, batch integration was performed using Harmony (version 1.0) (*Korsunsky et al., 2019*). Cell type was annotated according to the metadata from published datasets (*Zhong et al., 2020*; *Emont et al., 2022*).

### Animals study design

*Adipoq$^{Cre}$:Rosa26$^{LSL-tdTomato}$* (Adipoq:Td) mice were generated by breeding *Rosa26-LSL-tdTomato* (*Madisen et al., 2010*) mice with *Adipoq$^{Cre}$* mice (*Eguchi et al., 2011*). To generate *Csf1* CKO$^{Adipoq}$ mice, we first bred *Adipoq$^{Cre}$* with *Csf1$^{flox/flox}$* mice (*Harris et al., 2012*) to obtain *Adipoq$^{Cre}$ Csf1$^{flox/+}$*, which were then crossed with *Csf1$^{flox/flox}$* to generate *Csf1* CKO$^{Adipoq}$ mice and control (*Csf1$^{flox/flox}$*, *Csf1$^{flox/+}$*) siblings. *Csf1* CKO$^{Prrx1}$ mice were generated by a similar breeding strategy using *Prrx1$^{Cre}$* (*Logan et al., 2002*). All mouse lines, except *Csf1$^{flox/flox}$*, were obtained from Jackson Laboratory (Bar Harbor, ME, USA). For LPS-induced bone destruction, 6-week-old male mice were injected with 25 mg/kg LPS (Sigma Aldrich, St. Louis, MO) or PBS above calvariae. After 7 days, calvariae were collected and analyzed by microCT followed by H&E and TRAP staining.

### Micro-computed tomography (microCT) analysis

MicroCT analysis (microCT 35, Scanco Medical AG, Brüttisellen, Switzerland) was performed at 6 µm isotropic voxel size as described previously (*Chandra et al., 2017*). Briefly, the distal end of femur corresponding to a region at 0–2.8 mm below the growth plate was scanned at 6 µm isotropic voxel size to acquire a total of 466 µCT slices per scan. The images of the secondary spongiosa regions (0.6–1.8 mm below the lowest point of the growth plate) were contoured for trabecular bone analysis. At the femur midshaft, a total of 100 slices located 4.8–5.4 mm away from the distal growth plate were acquired for cortical bone analyses by visually drawing the volume of interest (VOI). In vertebrae, the region 50 slices away from the top and bottom end plates (~300 slices) was acquired for trabecular bone analysis. The trabecular bone within VOI was segmented from soft tissue using a threshold of 487.0 mgHA/cm$^3$ and a Gaussian noise filter (sigma = 1.2, support = 2.0). The cortical bone tissue was segmented using a threshold of 661.6 mgHA/cm$^3$ and a Gaussian noise filter (sigma = 1.2, support = 2.0). Three-dimensional standard microstructural analysis was performed to determine the geometric trabecular bone volume fraction (BV/TV), bone mineral density (BMD), trabecular thickness (Tb.Th), trabecular separation (Tb.Sp), trabecular number (Tb.N), and structure model index (SMI). For analysis of cortical bone, periosteal perimeter (Ps.Pm), endosteal perimeter (Ec.Pm), cortical bone area (Ct. Ar), cortical thickness (Ct.Th), and tissue mineral density (TMD) were recorded. All calculations were performed based on 3D standard microstructural analysis (*Bouxsein et al., 2010*).

Calvariae were scanned at 15 μm isotropic voxel size. The three-dimensional images were reconstructed to visualize the destructive area. A square region of 8 mm x 8 mm centered at the midline suture was selected for further quantitative analysis by ImageJ.

## Histology and bone histomorphometry

To obtain whole mount sections for immunofluorescent imaging of *Csf1* CKO*Adipoq* mouse bones, freshly dissected femurs were fixed in 4% PFA for 1 day, decalcified in 10% EDTA for 4–5 days, and then immersed into 20% sucrose and 2% polyvinylpyrrolidone (PVP) at 4 °C overnight. Then samples were embedded into medium containing 8% gelatin, 20% sucrose and 2% PVP and sectioned at 50 μm in thickness. Sections were incubated with rat anti-Endomucin (Santa cruz, sc-65495), rabbit anti-osterix (Abcam, ab22552), rat anti-F4/80 (Biolegend, 123101), or rabbit anti-Perilipin (Cell signaling, 9349) at 4 °C overnight followed by Alexa Fluor 488 anti-rat (Abcam, ab150155) or Alexa Fluor 647 anti-rabbit (Abcam, ab150075) secondary antibodies incubation 1 hr at RT.

To obtain paraffin sections, mouse calvarial bones were fixed in 4% PFA for 24 hr and decalcified in 10% EDTA for 5–7 days at 4 °C. Samples were then embedded in paraffin, sectioned at 6 μm in thickness, and processed for H&E staining and tartrate-resistant acid phosphatase (TRAP) staining (Acid Phosphatase, Leukocyte (TRAP) Kit, Sigma-Aldrich, 387 A).

To obtain cryosections without decalcification, mouse bones were dissected and fixed in 4% PFA for 24 hr, dehydrated in 30% sucrose in PBS, embedded in optimal cutting temperature (OCT) compound, and sectioned at 6 μm in thickness using a cryofilm tape (Section Lab, Hiroshima, Japan). Fluorescent TRAP staining was performed as described previously (*Dyment et al., 2015*). To measure dynamic histomorphometry, mice received calcein (10 mg/kg, Sigma Aldrich) and xylenol orange (90 mg/kg, Sigma Aldrich) at 9 and 2 days, respectively, before euthanization. Sagittal cryosections of tibiae prepared with cryofilm tape were used for dynamic histomorphometry. Sections were scanned by a Nikon Eclipse 90i fluorescence microscope and areas within the secondary spongiosa were quantified by OsteoMeasure Software (OsterMetrics, Decatur, GA, USA). The primary indices include total tissue area (TV), trabecular bone perimeter (BS), single- and double-labeled surface, and interlabel width. Mineralizing surface (MS), bone formation rate (BFR), and surface-referent bone formation rate (BFR/BS, $\mu m^3/\mu m^2/d$) were calculated as described by *Dempster et al., 2013*.

## Mechanical testing

Femurs were subjected to three-point bending tests in a universal test frame (5542, Instron, Norwood, MA, USA) equipped with a 100 N load cell. Samples were tested in a custom-built aluminum fixture, with a span of 10 mm between centerlines of 3 mm diameter stainless steel cylindrical rollers. Samples were aligned such that support beams were under both metaphyseal regions and loads were applied at the mid-point in an anteroposterior direction, perpendicular to the long axis of the femur. Tests were performed in displacement control, at a rate of 1.8 mm/min until failure and output data was recorded at 100 Hz. Force-displacement data was used to determine peak force (N), stiffness (N/mm), and work to failure (J).

## Hematopoietic phenotyping of bone marrow cells

Peripheral blood of mice was collected retro-orbitally for complete blood count (CBC) measurement using a Hemavet 950 (Drew Scientific, Inc). Bone marrow was flushed from femurs of 3-month-old control and *Csf1* CKO*Adipoq* mice and pre-treated with Fc-blocker (Invitrogen, 14-0161-81). The HSPC compartment was analyzed by staining for lineage markers biotin-Ter-119, -Mac-1, -Gr-1, -CD4, -CD8α, -CD5, -CD19 and -B220 (eBioscience, 13-5921-85, 13-0051-85, 13-5931-86, 13-0112-86, 13-0452-86, 13-0041-86, 13-0081-86, 13-0193-86) followed by staining with streptavidin-PE-Texas Red (Invitrogen, SA1017), rat anti-mouse cKit APC-Cy7 (eBioscience, 47-1171-82), rat anti-mouse Sca1 PerCP-Cy5.5 (eBioscience, 45-5981-82), hamster anti-mouse CD48 APC (eBioscience, 17-0481-82) and rat anti-mouse CD150 PE-Cy7 (Biolegend, 115914). To examine macrophages, flushed bone marrow cells were stained with rat anti-mouse F4/80 BV711 (Biolegend, 123147) and rat anti-mouse CD11b FITC (Biolegend, 101205). To analyze additional hematopoietic subpopulations, we stained bone marrow cells with rat anti-mouse CD45 Alexa Fluor 700 (Biolegend, 147715), rat anti-mouse Ly-6G APC (BioLegend, 127613), rat anti-mouse CD170 PE (Siglec-F) (Biolegend, 155503), rat anti-mouse Ly-6C PerCP (BioLegend, 128027), rat Anti-CD11b BV605 (BD science, 563015), rat anti-mouse/human

CD45R/B220 PerCP (Biolegend, 103233), rat anti-mouse CD3 FITC (Biolegend, 100203), and rat anti-mouse TER-119/Erythroid Cells APC (Biolegend, 116211) antibodies. All flow cytometry experiments were performed by either a LSR A or a BD LSR Fortessa flow cytometer and analyzed by FlowJo v10.5.3.

## Cell culture

For in vitro osteoclastogenesis, bone marrow cells flushed from long bones were added to dishes for obtaining bone marrow macrophages (BMMs) as described previously (*Ng et al., 2019*; *Yuan et al., 2015*). BMMs were then seeded at $1 \times 10^6$ cells/well in 24-well plates and stimulated with 30 ng/mL RANKL and 30 ng/mL M-CSF (R&D Systems, Minneapolis, MN, USA) for 5 days to generate mature OCs. TRAP staining was performed using a TRAP kit (Sigma Aldrich, 387 A). Osteoclasts were quantified by counting the number of TRAP+, multinucleated cells (>3nuclei/cell) per well.

## ELISA assays

To measure IL-34 amount, mouse tibiae and femurs were cut at both ends, vertically placed in a centrifuge tube, and briefly centrifuged. The pelleted bone marrow was lysed in RIPA buffer and analyzed using Mouse IL-34 Quantikine ELISA Kit (R&D Systems, M3400).

Sera were collected during mouse euthanization for measuring bone turnover markers, collagen type I C-telopeptide degradation products (mouse CTX-I ELISA Kit, AFG Scientific, Northbrook, IL, USA) and N-terminal propeptide of type I procollagen (Immunotag Mouse PINP ELISA Kit, G-Bioscience, St. Louis, MO, USA) respectively, according to the manufacturer's instructions.

## RNA analysis

Sorted cells or bone marrow centrifuged from long bones was collected in Tri Reagent (Sigma Aldrich) for RNA purification. A Taqman Reverse Transcription Kit (Applied BioSystems, Inc, Foster City, CA, USA) was used to reverse transcribe mRNA into cDNA. The power SYBR Green PCR Master Mix Kit (Applied BioSystems, Inc) was used for quantitative real-time PCR (qRT-PCR). Primers for *Csf1* gene are 5'-ATGAGCAGGAGTATTGCCAAGG-3' (forward) and 5'- TCCATTCCCAATCATGTGGCTA-3' (reverse), and primes for β-actin gene are 5'-GGCTGTATTCCCCTCCATCG-3'(forward) and 5'-CCAG TTGGTAACAATGCCATGT-3' (reverse).

## Statistical analyses

Data are expressed as means ± standard deviation (SD). For comparisons between two groups, unpaired two-sample student's t-test was applied. For comparisons amongst multiple groups across two fixed effect factors (e.g. genotype and age), two-way ANOVA was applied, followed by Tukey-Kramer multiple comparison test to account for family-wise type I error using Prism 8 software (GraphPad Software, San Diego, CA). In all tests, the significance level was set at $\alpha$=0.05. For assays using primary cells, experiments were repeated independently at least three times and representative data were shown here. Values of p<0.05 were considered statistically significant.

## Acknowledgements

We thank MicroCT Imaging Core and Biomechanics Core at Penn Center for Musculoskeletal Disorders (PCMD) for their assistance with microCT analysis and mechanical testing, respectively. This study was supported by NIH grants NIA R01AG069401, NIAMS R21AR078650 (to LQ), R00AR067283 (to ND), P30AR069619 (to PCMD), T32HL007439 (to SB), R01AG045040 (to JXJ) and Welch Foundation grant (AQ-1507) (to JXJ).

## Additional information

### Competing interests

Jean X Jiang: Reviewing editor, eLife. The other authors declare that no competing interests exist.

## Funding

| Funder | Grant reference number | Author |
| --- | --- | --- |
| National Institute on Aging | R01AG069401 | Ling Qin<br>Jean X Jiang |
| National Institute of Arthritis and Musculoskeletal and Skin Diseases | R21AR078650 | Ling Qin |
| Welch Foundation | AQ-1507 | Jean X Jiang |
| National Institute of Arthritis and Musculoskeletal and Skin Diseases | R00AR067283 | Nathanial Dyment |
| National Institute on Aging | R01AG045040 | Jean X Jiang |
| National Heart, Lung, and Blood Institute | T32HL007439 | Shovik Bandyopadhyay |

The funders had no role in study design, data collection and interpretation, or the decision to submit the work for publication.

## Author contributions

Leilei Zhong, Conceptualization, Data curation, Formal analysis, Validation, Investigation, Methodology, Writing - review and editing; Jiawei Lu, Jiankang Fang, Formal analysis, Validation, Investigation, Visualization, Methodology, Writing - review and editing; Lutian Yao, Data curation, Software, Formal analysis, Investigation, Visualization; Wei Yu, Tao Gui, Nicholas Holdreith, Catherine A Bautista, Investigation; Michael Duffy, Methodology, Writing - review and editing; Xiaobin Huang, Formal analysis; Shovik Bandyopadhyay, Data curation; Kai Tan, Chider Chen, Formal analysis, Methodology; Yongwon Choi, Conceptualization; Jean X Jiang, Resources; Shuying Yang, Wei Tong, Resources, Supervision, Writing - review and editing; Nathanial Dyment, Resources, Supervision; Ling Qin, Conceptualization, Resources, Formal analysis, Supervision, Funding acquisition, Validation, Investigation, Visualization, Methodology, Writing - original draft, Project administration, Writing - review and editing

## Author ORCIDs

Chider Chen ⓘ http://orcid.org/0000-0003-2899-1208
Jean X Jiang ⓘ http://orcid.org/0000-0002-2185-5716
Shuying Yang ⓘ http://orcid.org/0000-0002-7126-6901
Nathanial Dyment ⓘ http://orcid.org/0000-0001-8708-112X
Ling Qin ⓘ http://orcid.org/0000-0002-2582-0078

## Ethics

All animal work performed in this report was approved by the Institutional Animal Care and Use Committee (IACUC) at the University of Pennsylvania under Protocols 806887 and 804112. University Laboratory Animal Resources (ULAR) of the University of Pennsylvania is responsible for the procurement, care, and use of all university-owned animals as approved by IACUC. Animal facilities in the University of Pennsylvania meet federal, state, and local guidelines for laboratory animal care and are accredited by the Association for the Assessment and Accreditation of Laboratory Animal Care International.

## Decision letter and Author response

Decision letter https://doi.org/10.7554/eLife.82112.sa1
Author response https://doi.org/10.7554/eLife.82112.sa2

# Additional files

## Supplementary files
• MDAR checklist

## Data availability

Pre-aligned scRNA-seq matrix files were acquired from previously published dataset GEO GSE145477 and snRNA-seq matrix files were from GSE176171 (mouse), human scRNA-seq matrix files were acquired from EMBL-EBI E-MTAB-9139 (human). All data are available as source data files with submission.

The following previously published datasets were used:

| Author(s) | Year | Dataset title | Dataset URL | Database and Identifier |
|---|---|---|---|---|
| Zhong et al | 2020 | Single cell transcriptomics identifies a unique adipose lineage cell population that regulates bone marrow environment | https://www.ncbi.nlm.nih.gov/geo/query/acc.cgi?acc=GSE145477 | NCBI Gene Expression Omnibus, GSE145477 |
| Emont et al | 2022 | A single-cell atlas of human and mouse white adipose tissue | https://www.ncbi.nlm.nih.gov/geo/query/acc.cgi?acc=GSE176171 | NCBI Gene Expression Omnibus, GSE176171 |
| de Jong et al | 2021 | The multiple myeloma microenvironment is defined by an inflammatory stromal cell landscape. | https://www.ebi.ac.uk/biostudies/arrayexpress/studies/E-MTAB-9139 | EMBL-EBI, E-MTAB-9139 |

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
