## [Editor Report]

This fundamental work advances our understanding of bone marrow adipogenic lineage precursors as a major source of Csf1 in bone and important regulator of bone remodeling. The evidence supporting the conclusion is compelling, using Adipoq-Cre-based conditional deletion of Csf1 and scRNA-seq approaches. This paper is of potential interest to scientists who study bone marrow stem/progenitor cells, bone remodeling, and metabolism.

---

## [Decision Letter]

**Decision letter after peer review:**

Thank you for submitting your article "Csf1 from marrow adipogenic precursors is required for osteoclast formation and hematopoiesis in bone" for consideration by *eLife*. Your article has been reviewed by 2 peer reviewers, and the evaluation has been overseen by a Reviewing Editor and Mone Zaidi as the Senior Editor. The reviewers have opted to remain anonymous.

Essential revisions:

This is an important and well-executed set of study. The evidence provided generally support the main conclusion. The authors should consider to address several concerns raised by the reviewers. Particularly, it is necessary for the authors to perform (1) mechanical testing on bones from the cKO lines; (2) further characterization of molecular changes other than M-CSF in MALPs from the cKO lines; (3) characterization of the calvaria cells that Adipoq-Cre marks; and (4) more in-depth analyses on bone marrow hematopoiesis in Csf1 cKO mice. In addition, the authors should consider addressing several other concerns raised by the reviewers.

*Reviewer #1 (Recommendations for the authors):*

Congratulations on making this discovery from scRNA-seq datasets and validating it in multiple animal models. The functional studies indicate that MALPs/AdipoQ+ cells in the bone marrow produce functionally significant amounts of Csf1 and this is one means by which they have an outsized effect on bone remodeling. There are many strengths to the data and presentation, including use of two CKO lines, examination of both long bone and vertebral bone mass as several ages, employing the LPS model of osteolysis, and studying the effect of Csf1 deletion in MALPs on hematopoiesis and vasculature. I have a few questions and suggestions for improving the study and manuscript.

1) What are the controls for the CKO mice? They are labeled as WT in the Methods and figures. Are they WT or Csf1fl/fl mice? Or Cre+ mice? If they are one of the latter two and not WT, they can be labeled as controls. If they are WT, it should be explained why they are used, rather than other controls.

2) Do MALPS have to be in contact with osteoclast progenitors to stimulate osteoclast differentiation in vitro or do they secrete soluble M-CSF?

3) Osteopetrotic bones are more fragile. Have you performed mechanical testing on your CKO lines to determine their strength and stiffness?

4) Have you done additional transcriptomic studies on the Csf1 CKOAdipoQ mice? For example, have you performed a scRNA-seq or even bulk RNA-seq analysis on the bone marrow of Csf1∆AdipoQ mice and controls to examine transcriptomic changes other than M-CSF in these animals?

5) IL-34 has redundant roles with M-CSF. Does its expression change in Csf1 CKOAdipoQ mice? What about other cytokines and transcripts?

6) To what extent do you believe that the results in the osteolysis model are due to the reduction in macrophages that would be activated by LPS?

7) Do MALPs exist in humans? If so, do human MALPs produce M-CSF?

*Reviewer #2 (Recommendations for the authors):*

1. Abstract:

– The conclusion of the abstract ("Taken together…") needs to be more specific. The current sentence lacks the most important piece of information of this study, Csf1.

2. Introduction:

– The introduction is generally speaking well-written, with relevant literature comprehensively covered. The authors rightfully emphasize the importance of Csf1 derived from various types of mesenchymal cells including osteoblasts.

– I wonder which isoform(s) are predominantly expressed by MALPs. Could this be determined using immunohistochemical approaches?

3. Results:

– It is not entirely clear how the authors determine that Adipoq-Cre marks only MALP, but not LCP or EMP. Is it possible that MALPs may not stay entirely as adipoprogenitors and drift toward osteoblasts or their precursor cells?

– The authors determined that lipid-laden adipocytes do not express Csf1 based on publicly available snRNA-seq dataset from white adipose tissues. It is still possible that bone marrow adipocytes express Csf1, as they are known to be very different from white adipocytes.

– Can the authors discuss further why they did not detect any structural changes in vertebral trabecular bone?

4. Discussion:

– It would be helpful if the authors could also discuss the similarities and differences of bone phenotypes between Adipoq-Cre-driven M-CSF and RANKL CKO models.

---

## [Author Response]

Reviewer #1 (Recommendations for the authors):Congratulations on making this discovery from scRNA-seq datasets and validating it in multiple animal models. The functional studies indicate that MALPs/AdipoQ+ cells in the bone marrow produce functionally significant amounts of Csf1 and this is one means by which they have an outsized effect on bone remodeling. There are many strengths to the data and presentation, including use of two CKO lines, examination of both long bone and vertebral bone mass as several ages, employing the LPS model of osteolysis, and studying the effect of Csf1 deletion in MALPs on hematopoiesis and vasculature. I have a few questions and suggestions for improving the study and manuscript.1) What are the controls for the CKO mice? They are labeled as WT in the Methods and figures. Are they WT or Csf1fl/fl mice? Or Cre+ mice? If they are one of the latter two and not WT, they can be labeled as controls. If they are WT, it should be explained why they are used, rather than other controls.

As stated in Materials and methods, control mice are *Csf1^flox/flox^* and *Csf1^flox/+^* mice (both *Adipoq-Cre* negative) and they are siblings of *Csf1 CKO* mice (*Adipoq-Cre* positive). We bred *Adipoq-Cre Csf1^flox/+^* mice with *Csf1^flox/flox^* mice to generate *AdipoqCre Csf1^flox/flox^ (CKO), Adipoq-Cre Csf1^flox/+^ (Het), Csf1^flox/flox^ (WT)* and *Csf1^flox/+^ (WT) mice.* This breeding strategy does not produce Cre only sibling mice. To clarify this, we added the following explanation to Results:

“They grew normally with comparable body weight, femoral length, and tooth eruption as *Csf1^flox/flox^* and *Csf1^flox/+^* siblings, which are considered *WT* in this study (Figure 2A-C).”

2) Do MALPS have to be in contact with osteoclast progenitors to stimulate osteoclast differentiation in vitro or do they secrete soluble M-CSF?

Since there is no good marker to label osteoclast progenitors in vivo, we cannot directly address the first question. However, our previous studies showed that MALPs form an extensive network in the bone marrow and contact nearby bone marrow cells and bone surface cells, including osteoclasts, via their extensive cell processes (1, 2). Therefore, it is reasonable to assume that MALPs are in direct contact with osteoclast progenitors.

For the second question, Csf1 is normally expressed as a secreted glycoprotein, a secreted proteoglycan, or a membrane-spanning cell surface glycoprotein (3). All forms are functional. In the co-submitted manuscript by Dr. Baohong Zhao’s group (4), IHC shows Csf1 staining in MALPs (Author response image 1), suggesting that MALPs do express the membrane form of Csf1. Since MALPs do not proliferate in vitro (1), it is impossible to collect conditioned medium from these cells for ELISA analysis of Csf1 protein. In our rebuttal to Reviewer 2 Introduction comment, we also explain why we cannot use immunostaining approach to differentiate membrane bound Csf1 and soluble Csf1. Thus, we are unable to answer whether MALPs secrete soluble Csf1.

**Author response image 1. sa2fig1:** Adipoq+ cells in mouse bone marrow express Csf1 at the protein level. Fluorescent staining of Csf1 was performed on the long bones of *Adipoq-Cre mTmG* mice. Arrows point to GFP+Csf1+ cells. It was observed that the majority of bone marrow AdipoQ-expressing progenitor cells express Csf1 (1865 cells out of 2001 cells counted, n=3 mice, 93.2%). Adopted from ref (4).

3) Osteopetrotic bones are more fragile. Have you performed mechanical testing on your CKO lines to determine their strength and stiffness?

In 2015, JBMR published guidelines for bone mechanical testing (5). This review article states that mechanical properties are assessed at two length scales: the whole-bone (“structural”) level, eg, the femoral diaphysis; and the tissue (“material”) level, namely cortical bone. The primary difference between mechanical properties at these two length scales is that whole-bone properties are measured using intact bones and depend on bone size, whereas tissue-level properties assess the material and are size-independent. Thus, mechanical testing is mainly performed on femoral cortical bone, but not on the trabecular bone. In our study, femoral cortical bone structure and tissue mineral density (TMD) was not altered in *Csf1 CKO^Adipoq^* mice, predicting that their mechanical properties should remain the same as *WT* mice. To validate this, we performed 3-point bending analysis on femurs of *Csf1 CKO^Adipoq^* mice and did not detect any obvious changes in their mechanical properties, which is in line with normal cortical bone structure measured by microCT. These data are now included as Figure 2—figure supplement 2C. Accordingly, we modified our conclusion in Results to this sentence:

“Meanwhile, femoral cortical bone structure, tissue mineral density (TMD), and mechanical properties were not altered (Figure 2—figure supplement 2).”

4) Have you done additional transcriptomic studies on the Csf1 CKOAdipoQ mice? For example, have you performed a scRNA-seq or even bulk RNA-seq analysis on the bone marrow of Csf1∆AdipoQ mice and controls to examine transcriptomic changes other than M-CSF in these animals?

We thank the Reviewer for this suggestion. Since MALPs only constitute a small portion of bone marrow cells (~0.5%), sorting MALPs from *WT* and *Csf1 CKO^Adipoq^* mice for scRNA-seq or bulk RNA-seq requires a large cohort of animals, which is not only time consuming (> 6 months to accumulate enough mice and another 3 months for sorting and sequencing if everything goes smoothly), but also very expensive (animals, flow sorting, library construction, and sequencing). Instead, we analyzed bone marrow cytokine amount to address this comment. Using a mouse cytokine array kit with 111 cytokine antibodies on the membrane, we found no significant difference between the bone marrow of *WT* and *CKO* mice (Author response image 2). Thus, we believe that *Csf1* depletion does not drastically altered transcriptomics in MALPs. Indeed, these data were expected because MALPs do not express *Csf1r* and thus are not recipients for Csf1-Csf1R signaling. We acknowledge that this cytokine array assay is semi-quantitative and limited in the number of proteins to be analyzed. However, the main goal of this manuscript is to depict the role of MALP-derived Csf1 in bone remodeling and hematopoiesis. While thorough transcriptomics analysis of MALPs is interesting, we do not believe that this result will affect the overall conclusion.

**Author response image 2. sa2fig2:** Csf1 depletion in MALPs does not alter bone marrow cytokine expression. Bone marrow from 3-5-month-old *WT* and *Csf1 CKO^Adipoq^* mice (n=5/group) were centrifuged out from long bones, lysed by RIPA buffer, and subjected to cytokine array analysis using Mouse XL Cytokine Array Kit (Cat# ARY028, R&D Systems). Bone marrow from 2-3 mice was pooled for one membrane as indicated.

5) IL-34 has redundant roles with M-CSF. Does its expression change in Csf1 CKOAdipoQ mice? What about other cytokines and transcripts?

Using an Il-34 ELISA kit, we analyzed the amount of Il34 in the bone marrow and found no difference between *WT* (1.27 ± 0.44 µg/ml, n=6 mice) and *CKO* (1.08± 0.55 pg/ml, n=5 mice). As shown in Author response image 2, we also did not detect any significant changes in 111 cytokines using the mouse cytokine array kit. Accordingly, we add these sentences to the Results:

“Binding to the same receptor Csf1R, Il34 shares a redundant role with Csf1 (6). However, ELISA analysis found no change in Il34 amount in the *CKO* bone marrow (*WT* 1.27 ± 0.44 µg/ml vs *CKO* 1.08± 0.55 pg/ml, n=5-6 mice), suggesting no compensatory increase of Il34 in *CKO*.”

6) To what extent do you believe that the results in the osteolysis model are due to the reduction in macrophages that would be activated by LPS?

LPS is widely recognized as a potent activator of monocytes/macrophages (7). It causes acute inflammatory response by triggering the release of a vast number of inflammatory cytokines in various cell types (8). Thus, we do believe that LPS activates macrophages in *WT* calvariae. Using F4/80 staining, we found that the increase of macrophages in *WT* calvariae after LPS injection is subdued in *CKO* calvariae (Author response image 3). Whether these macrophages later become osteoclasts is not known. To focus on LPS-induced osteoclastogenesis, we prefer not to include these data in our manuscript.

**Author response image 3. sa2fig3:** 4/80 staining (brown) of mouse calvarial bone marrow at day 7 post a vehicle or an LPS injection.

7) Do MALPs exist in humans? If so, do human MALPs produce M-CSF?

Yes. We examined a recently published bone marrow scRNA-seq dataset from human patients with multiple myeloma (9) and found strong *Csf1* expression in Adipoq+ cells from control bone marrow, which was obtained by sternal aspiration from donors undergoing cardiothoracic surgery or by manual bone marrow collection from femur heads collected after hip replacement surgery. As shown in the newly added Figure 1H, most mesenchymal lineage cells express both *Adipoq* and *Csf1*. Note that in this paper, bone marrow cells were collected by aspiration, followed by freezing and thaw, cell sorting, and scRNA-seq analysis. It seems that most cells in their dataset are MALPs. We recently harvested human bone marrow from femur heads using an enzymatic digestion method. There are more distinct cell clusters, particularly osteogenic cells with no adipogenic marker expression. In our dataset, we also observe high *Csf1* expression in the Adipoq+ cell cluster. We plan to publish our human scRNA-seq data in the near future.

Reviewer #2 (Recommendations for the authors):1. Abstract:– The conclusion of the abstract ("Taken together…") needs to be more specific. The current sentence lacks the most important piece of information of this study, Csf1.

This conclusion is now modified to include Csf1:

“Taken together, our studies demonstrate that MALPs synthesize Csf1 to control bone remodeling and hematopoiesis.”

2. Introduction:– The introduction is generally speaking well-written, with relevant literature comprehensively covered. The authors rightfully emphasize the importance of Csf1 derived from various types of mesenchymal cells including osteoblasts.– I wonder which isoform(s) are predominantly expressed by MALPs. Could this be determined using immunohistochemical approaches?

We thank the Reviewer for his/her approval of Introduction.

The CSF-1 isoforms are synthesized from a full-length and a truncated precursor (11). The Nterminal 150 amino acids of both precursors are identical and sufficient for in vitro biological activity. The remainder of the precursor sequences determines how the isoforms are processed and expressed. The secreted isoforms are derived by differential proteolysis from the full-length precursor in the secretory vesicle: the secreted glycoprotein by N-terminal cleavage; and the secreted proteoglycan, on which the 18-kDa chondroitin sulfate chains (hexamers) remain attached, by C-terminal cleavage. The cell-surface isoform is derived from the truncated precursor. This precursor is encoded by an alternatively spliced mRNA in which the regions encoding the proteolytic cleavage and glycosaminoglycan addition sites are spliced out. Thus, this uncleaved precursor is expressed stably on the cell surface as a membrane-spanning glycoprotein. Unfortunately, commercially available Csf1 antibodies are mostly generated against *E. coli*-derived recombinant mouse M-CSF containing the common N-terminal region. For example, Csf1 antibody from R&D systems (MAB4161) uses Lys33Glu262 as immunogen. As such designed, they cannot differentiate Csf1isoforms. Thus, we are unable to use IHC or western blot to study which isoforms are predominantly expressed by MALPs.

3. Results:– It is not entirely clear how the authors determine that Adipoq-Cre marks only MALP, but not LCP or EMP. Is it possible that MALPs may not stay entirely as adipoprogenitors and drift toward osteoblasts or their precursor cells?

In our previous publications (1, 2), we have performed thorough analyses demonstrating that *Adipoq-Cre* marks only MALPs but not osteoblasts and osteocytes in *Adipoq-Cre Tomato 2.3kbCol1a1-GFP* mice up to 6 months of age. Moreover, we found that *Csf1* expression in *Csf1 CKO^Adipoq^* mice is significantly decreased in the bone marrow, but not altered in the cortical bone, a region made of osteocytes (Figure 2D). Taken together, we are confident that the bone effect observed in *Csf1 CKO^Adipoq^* mice here is due to a lack of Csf1 in MALPs but not in osteogenic cells nor in bipotent mesenchymal progenitors.

– The authors determined that lipid-laden adipocytes do not express Csf1 based on publicly available snRNA-seq dataset from white adipose tissues. It is still possible that bone marrow adipocytes express Csf1, as they are known to be very different from white adipocytes.

In the co-submitted manuscript by Dr. Baohong Zhao’s group (4), qRT-PCR and IHC analyses clearly show that Csf1 is expressed in MALPs but not in lipid-laden adipocytes (LiLAs) (Author response image 1 and Author response image 4).

**Author response image 4. sa2fig4:** Csf1 is expressed in MALPs but not LiLAs in bone marrow. (A) qRT-PCR analysis of Csf1 expression in bone marrow adipocytes BMAds, which are LiLAs in our terminology, and bone marrow Adipoq-lineage progenitors, which are MALPs in our terminology, sorted from the bone marrow of *Adipoq-Cre mTmG* mice. (B) Fluorescent staining of Perillipin and Csf1 was performed on mouse long bones. Csf1 expression was not detected in mature bone marrow adipocytes (Perilipin1+) (0 cells out of 115 cells counted, n=3 mice, 0%). Adopted from ref (4).

– Can the authors discuss further why they did not detect any structural changes in vertebral trabecular bone?

We hope that the following paragraph in the Discussion is sufficient to address this comment:

“To our surprise, while high trabecular bone mass was evident in long bones, not changes were detected in the vertebrae of *Csf1 CKO^Adipoq^* mice. Based on our scRNA-seq data, *Adipoq* expression is highly specific for MALPs. Since Csf1 is also expressed at a considerable level in LCPs, we believe that LCP-derived Csf1, which should not be affected in *Csf1 CKO^Adipoq^* mice, might be sufficient for promoting osteoclastogenesis in vertebrae.”

4. Discussion:– It would be helpful if the authors could also discuss the similarities and differences of bone phenotypes between Adipoq-Cre-driven M-CSF and RANKL CKO models.

In our first submission, we have discussed the similarities and differences of phenotypes between *Csf1 CKO^Adipoq^* mice and *Rankl CKO^Adipoq^* mice at difference places in the Discussion. Here are some examples:

“In our last report, we demonstrated that RANKL from MALPs promotes osteoclast formation and bone loss under normal and diseased conditions (2). In this study, we found that Csf1 from MALPs shares similar action in regulating bone resorption.”

“In our previous study of mice with RANKL deficiency in MALPs (*Rankl CKO^Adipoq^* mice), we proposed that osteoclast formation is controlled by various mesenchymal subpopulations in a site-dependent fashion. Our current research further substantiates this conclusion. We found that *Csf1 CKO^Adipoq^* mice, similar to *Rankl CKO^Adipoq^* mice, exhibit the trabecular bone phenotype but have normal long bone growth and cortical bone.”

“In this study, we found that femoral trabecular bone mass increased in *Csf1 CKO^Adipoq^* mice by ~73% at 3 months of age, comparable to the trabecular bone mass increase in *Csf1 CKO^Dmp1^* mice at 4 months of age (12). In our previous study, we found that *Rankl CKO^Adipoq^* mice develop osteopetrosis at 1 month of age while *Rankl CKO^Dmp1^* mice at the same age have normal bone mass (2).”

“Unlike RANKL, which is highly specific for mature osteoclast formation, Csf1 has much broader actions in tissue development, homeostasis, and repair (13). Our previous study with *Rankl CKO^Adipoq^* mice mainly detected bone phenotypes due to suppressed osteoclast formation. We did not notice obvious changes in bone marrow cellularity in those mice. However, *Csf1 CKO^Adipoq^* mice have hematopoietic phenotypes.”

To add further comparison, we write the following sentence in the Discussion to compare the vertebrae bone phenotype:

“On the contrary, while *Rankl* shares a similar expression pattern as *Csf1* in the scRNA-seq data, *Rankl CKO^Adipoq^* mice show a strong high trabecular bone mass phenotype in the vertebrae.”

References

Zhong L, Yao L, Tower RJ, Wei Y, Miao Z, Park J, et al. Single cell transcriptomics identifies a unique adipose lineage cell population that regulates bone marrow environment. *ELife.* 2020;9:e54695.

Yu W, Zhong L, Yao L, Wei Y, Gui T, Li Z, et al. Bone marrow adipogenic lineage precursors promote osteoclastogenesis in bone remodeling and pathologic bone loss. *J Clin Invest.* 2021;131(2):e140214.

Stanley ER. In: Oppenheim JJ, and Feldmann M eds. *Cytokine Reference: A Compendium of Cytokines and Other mediators of Host Defence* London, United Kingdom: Academic Press; 2000:911-34.

Inoue K, Xia Y, Qin Y, Jiang JX, Greenblatt MB, and Zhao B. Bone marrow AdipoQlineage progenitors are a major cellular source of M-CSF that dominates bone marrow macrophage development, osteoclastogenesis and bone mass. *Co-submitted manuscript.*

Jepsen KJ, Silva MJ, Vashishth D, Guo XE, and van der Meulen MC. Establishing biomechanical mechanisms in mouse models: practical guidelines for systematically evaluating phenotypic changes in the diaphyses of long bones. *J Bone Miner Res.* 2015;30(6):951-66.

Lelios I, Cansever D, Utz SG, Mildenberger W, Stifter SA, and Greter M. Emerging roles of IL-34 in health and disease. *J Exp Med.* 2020;217(3):e20190290.

Takashiba S, Van Dyke TE, Amar S, Murayama Y, Soskolne AW, and Shapira L. Differentiation of monocytes to macrophages primes cells for lipopolysaccharide stimulation via accumulation of cytoplasmic nuclear factor kappaB. *Infect Immun.* 1999;67(11):5573-8.

Ngkelo A, Meja K, Yeadon M, Adcock I, and Kirkham PA. LPS induced inflammatory responses in human peripheral blood mononuclear cells is mediated through NOX4 and GiÎ± dependent PI-3kinase signalling. *J Inflamm (Lond).* 2012;9(1):1.

de Jong MME, Kellermayer Z, Papazian N, Tahri S, Hofste Op Bruinink D, Hoogenboezem R, et al. The multiple myeloma microenvironment is defined by an inflammatory stromal cell landscape. *Nat Immunol.* 2021;22(6):769-80.

Logan M, Martin JF, Nagy A, Lobe C, Olson EN, and Tabin CJ. Expression of Cre

Recombinase in the developing mouse limb bud driven by a Prxl enhancer. *Genesis.* 2002;33(2):77-80.

Pixley FJ, and Stanley ER. CSF-1 regulation of the wandering macrophage: complexity in action. *Trends Cell Biol.* 2004;14(11):628-38.Werner SL, Sharma R, Woodruff K, Horn D, Harris SE, Gorin Y, et al. CSF-1 in Osteocytes Inhibits Nox4-mediated Oxidative Stress and Promotes Normal Bone Homeostasis. *JBMR Plus.* 2020;4(7):e10080.Sehgal A, Irvine KM, and Hume DA. Functions of macrophage colony-stimulating factor (CSF1) in development, homeostasis, and tissue repair. *Semin Immunol.* 2021;54:101509.